# Relationship between COVID-19 and ICU-Acquired Bloodstream Infections Related to Multidrug-Resistant Bacteria

**DOI:** 10.3390/antibiotics12071105

**Published:** 2023-06-25

**Authors:** Antoine Piantoni, Marion Houard, Gaetan Piga, Ghadi Zebian, Sarah Ruffier des Aimes, Bérénice Holik, Frédéric Wallet, Anahita Rouzé, Louis Kreitmann, Caroline Loiez, Julien Labreuche, Saad Nseir

**Affiliations:** 1CHU de Lille, Service de Médecine Intensive Réanimation, F-59000 Lille, France; 2CHU de Lille, Laboratoire de Bactériologie-Hygiène, Centre de Biologie Pathologie, F-59000 Lille, France; 3Inserm U1285, Université de Lille, CNRS, UMR 8576-UGSF, F-59000 Lille, France; 4CNRS, UMR 8576-UGSF, F-59000 Lille, France; 5Inserm, U1285, F-59000 Lille, France; 6Centre for Antimicrobial Optimisation, Department of Infectious Disease, Faculty of Medicine, Imperial College London, London W12 0HS, UK; 7ICU West, The Hammersmith Hospital, Du Cane Road, London W12 0HS, UK; 8Department of Biostatistics, CHU de Lille, F-59000 Lille, France

**Keywords:** SARS-CoV-2, COVID-19, bloodstream infection, ICU, multidrug-resistant bacteria, critical illness

## Abstract

A bloodstream infection (BSI) is a severe ICU-acquired infection. A growing proportion is caused by multidrug-resistant bacteria (MDRB). COVID-19 was reported to be associated with a high rate of secondary infections. However, there is a lack of data on the relationship between COVID-19 and the incidence of MDRB ICU-acquired BSI. The aim of this study was to evaluate the relationship between COVID-19 and ICU-acquired BSI related to MDRB. This retrospective study was conducted in a single-center ICU during a one-year period. All adult patients admitted for more than 48 h were included. The cumulative incidence of ICU-acquired BSI related to MDRB was estimated using the Kalbfleisch and Prentice method. The association of COVID-19 status with the risk of ICU-acquired BSI related to MDRB was assessed using cause-specific Cox’s proportional hazard model. Among the 1320 patients included in the analysis, 497 (37.65%) had COVID-19. ICU-acquired BSI related to MDRB occurred in 50 patients (36 COVID patients (7%) and 14 non-COVID patients (1.6%)). Extended-spectrum beta-lactamase Enterobacteriacae (46%) and carbapenem-resistant *Acinetobacter baumannii* (30%) were the most commonly isolated MDRB. COVID-19 was significantly associated with a higher risk of MDRB ICU-acquired BSI (adjusted cHR 2.65 (1.25 to 5.59) for the whole study period). However, this relationship was only significant for the period starting at day 15 after ICU admission. ICU-acquired BSI related to MDRB was significantly associated with ICU mortality (HR (95%CI) 1.73 (1–3)), although COVID-19 had no significant impact on this association (p het 0.94). COVID-19 is significantly associated with an increased risk of ICU-acquired BSI related to MDRB, mainly during the period starting at day 15 after ICU admission.

## 1. Introduction

Patients hospitalized in the ICU are at a high risk of infection because of exposure to many risk factors, such as invasive procedures, antibiotic treatment, acute illness, and other patient-related factors [1]. In the large EPIC III epidemiological international study, 54% of patients had an infection treated in the ICU, including 22% that were ICU-acquired [2]. Bloodstream infections (BSI) represented 15% of these ICU-acquired infections. A growing proportion of BSI are related to multidrug-resistant bacteria (MDRB), with a prevalence of difficult-to-treat MDRB of 23.5%, and panresistant MDRB of 1.5% [3]. BSI related to MDRB are associated with high mortality and increased morbidity. However, the mortality attributable to multidrug resistance is still a matter for debate [4].

“Severe acute respiratory syndrome coronavirus 2” (SARS-CoV-2) caused a pandemic starting from January 2020 and caused around 7 million deaths worldwide in February 2023 [5]. Acute respiratory distress syndrome (ARDS) linked to SARS-COV-2 required hospitalization in the ICU in 20% of hospitalized patients in 2020 [6,7].

Although bacterial co-infections are not common at hospital admission in COVID-19 patients [8,9,10], the risk of nosocomial infections is high, particularly in the ICU. For example, in a multicenter cohort of 4994 patients hospitalized in the ICU, Conway-Morris et al. [11] reported an incidence of ICU-acquired infections of approximately 54%, including 25% of infections related to MDRB. The main ICU-acquired infections reported are ventilator-acquired pneumonia [12,13,14], invasive pulmonary aspergillosis [15,16], and BSI [17,18]. A retrospective analysis of the EUROBACT II study also suggested higher mortality for BSI associated with SARS-CoV-2 compared to other BSI in the ICU [18].

Several factors could ex”plain the high incidence of ICU-acquired infections in COVID-19 patients, including long duration of ICU stay, prolonged duration of mechanical ventilation, increased exposure to antimicrobials, use of corticosteroids, immune dysfunction, and altered microbiota [19,20,21,22].

Whilst the incidence of ICU-acquired infections appears to be higher in COVID patients, as compared to non-COVID patients [23,24,25], little data are available on ICU-acquired BSI. Therefore, we conducted this retrospective analysis of prospectively collected data to compare the cumulative incidence of ICU-acquired BSI related to MDRB between COVID-19 and non-COVID-19 patients. The secondary objectives were to evaluate the effect of COVID-19 on the relationship between ICU-acquired BSI related to MDRB and mortality, duration of invasive mechanical ventilation (IMV), and length of ICU stay.

## 2. Patients and Methods

### 2.1. Population and Definitions

We performed a retrospective single-center study at the ICU of Lille university hospital in France. All adult patients hospitalized for >48 h between 1 January and 31 December 2020 were included.

SARS-CoV-2 infection was defined as a positive nasal or tracheal RT-PCR at baseline. ICU-related BSI was defined as positive blood culture occurring >48 h after admission. MDRB were defined as follows: extended spectrum beta-lactamase (ESBL) *Enterobacteriaceae*, carbapenem-resistant *Enterobacteriaceae*, methicillin-resistant *Staphylococcus aureus* (MRSA), *Pseudomonas aeruginosa* resistant to imipenem and ceftazidime, carbapenem-resistant *Acinetobacter baumannii*, and vancomycin-resistant *Enterococcus* (VRE). The bacteria were identified by matrix-assisted laser desorption ionization time of flight mass spectrometry (MALDI-TOF-MS) with a Microflex mass spectrometer (Bruker Daltonik S. A., Wissembourg, France) according to the manufacturer’s instructions after extraction using formic acid. The antimicrobial susceptibility tests were performed using the Vitek 2 system (BioMérieux, Marcy-l’Étoile, France), combined with the MASTDISCS ID ESBL detection disc diffusion tests (Mast Diagnostics, Amiens, France) to confirm the presence of ESBL or overproduction of cephalosporinase. In case of carbapenemase, the O.K.N.V.I. Resist Coris test (CorisBioconcept, Gembloux, Belgium) allowed us to determine the type of carbapenemase. The clinical breakpoints were interpreted using criteria proposed by the “Comité de l’Antibiogramme de la Société Française de Microbiologie” (CA-SFM EUCAST 2019).

### 2.2. Data Collection

Data, prospectively collected in an electronic medical file, were extracted into an electronic case report form, from the first day of admission until ICU discharge. Only first episodes of ICU-acquired BSI related to MDRB were taken into account.

Data collected at baseline included: age, gender, body mass index (BMI), dates of ICU admission and discharge, Simplified Acute Physiology Score (SAPS) II score [26], Sequential Organ Failure Assessment (SOFA) [27], COVID-19 status, comorbidities, recent hospitalization (<3 months before ICU admission), surgery, antibiotic treatment or MDRB colonization in the 3 months preceding ICU admission, type of admission (medical or surgical), location before ICU admission, and reason for ICU admission.

Data collected during ICU stay included: invasive procedures (central venous, arterial, dialysis catheters, IMV, tracheostomy, prone positioning, extracorporeal membrane oxygenation (ECMO)/extracorporeal life support (ECLS)), other treatments (parenteral nutrition, transfusion, antibiotics, steroids, tocilizumab, other immunosuppressive agents), MDRB colonization, and MDRB BSI.

### 2.3. Statistical Analysis

Patient characteristics (at admission or during ICU stay) were described according to COVID-19 status without statistical comparisons. Differences in patient’s characteristics according to COVID-19 status were evaluated using standardized differences and absolute values >20% were interpreted as meaningfully different. Categorical variables are reported as number and percentage, whereas quantitative variables are expressed as median with interquartile range (25th–75th percentile).

We estimated the cumulative incidence of ICU-acquired BSI related to MDRB using the Kalbfleisch and Prentice method [28] considering ICU discharge (alive or dead) as the competing event, by capping the length of ICU stay at day 60 since no MDRB occurred after 59 days. The association of COVID-19 status with the risk of ICU-acquired BSI related to MDRB was assessed using cause-specific Cox’s proportional hazard model regarding the causal research question, with ICU-acquired BSI related to MDRB taken as event of interest [29], before and after adjustment for pre-specified baseline confounders (age, SAPS II, immunosuppression [1], recent antibiotic treatment, colonization related to MDRB at ICU admission, and chronic dialysis). The cause-specific hazard ratio (cHR) for patients with COVID-19 vs. patients without COVID-19 was derived from Cox’s regression models with their 95% CIs as effect size. The proportional hazard assumption was assessed by using Schoenfeld residual plots and the Grambsch and Therneau test. Since the assumption was not satisfied, the effect of COVID-19 was further estimated for two separate periods (0- to 30-day vs. 31- to 60-day follow-up periods) by using a time-dependent coefficient in Cox’s regression models.

We investigated the association of ICU-acquired BSI related to MDRB with prognostic outcomes censored at day 60 (ICU mortality, IMV duration, length of ICU stay) by using univariable and multivariable Cox’s regression models with cause-specific hazards for ICU mortality (considering ICU death as an event of interest and ICU discharge alive as a competing event), IMV duration (considering extubation alive as an event of interest and death under IMV as a competing event) and length of ICU stay (considering ICU discharge alive as an event of interest and ICU mortality as a competing event), and by treating ICU-acquired BSI related to MDRB as time-varying covariate. Multivariable Cox’s models were performed to adjust the associations for pre-specified confounders known to be associated with prognostic outcomes (age, SAPS II, immunosuppression, recent hospitalization, recent antibiotic treatment, colonization related to MDRB at ICU admission, and chronic dialysis). In addition, subgroup analyses according to COVID-19 status were performed by separate Cox’s regressions models according to COVID-19 status given the strong deviation in proportional hazard assumption for COVID-19. Statistical testing was performed with a two-tailed α level of 0.05. Data were analyzed using the SAS software package, release 9.4 (SAS Institute, Cary, NC, USA).

## 3. Results

### 3.1. Patient Characteristics

A total of 1320 patients, admitted between 1 January and 31 December 2020, were included in this study. Among them, 497 (37.65%) patients had COVID-19. Fifty (3.79%) patients had ICU-acquired BSI related to MDRB, including 36 (7.24%) in the COVID-19 group, and 14 (1.70%) in the non-COVID-19 group.

Patients were mostly male, with a median age of between 60 and 65 years (Table 1). Median BMI was higher in the COVID-19 group than in the non-COVID-19 group. Severity scores were comparable between groups. Active smoking, alcohol consumption, chronic cardiac failure, chronic lung disease, recent hospitalization, surgery, prior antibiotic treatment, MDR colonization at ICU admission, and all types of immunosuppression were less common in the COVID-19 group, as compared with the non-COVID-19 group. Causes for admission were almost exclusively acute respiratory failure in the COVID-19 group, although other causes were found in the non-COVID-19 group.

During ICU stay, exposure to central venous catheters and dialysis catheters were comparable between COVID-19 and non-COVID-19 patients, whereas a higher percentage of arterial catheters, tracheostomy, prone positioning, and ECMO/ECLS was found in the COVID-19 group, as compared with the non-COVID-19 group. Antibiotic treatment, steroids, and parenteral nutrition were more common in the COVID-19 group, as compared with the non-COVID-19 group (Table 2).

### 3.2. Characteristics of Patients with ICU-Acquired BSI Related to MDRB According to COVID-19 Status

The bacteria responsible for the first episodes of ICU-acquired BSI related to MDRB were almost exclusively Gram-negative bacilli (Table 3). No VRE was isolated during the study period.

A higher proportion of carbapenem-resistant *Enterobacteriaceae*, carbapenem-resistant *Acinetobacter baumannii*, and multidrug-resistant *Pseudomonas aeruginosa* was found in the COVID-19 group than in the control group. All carbapenemase-producing strains were oxacillinase-48. The majority of BSI were of pulmonary or catheter-related origin, regardless of COVID-19 status.

### 3.3. Relationship between COVID-19 and ICU-Acquired BSI Related to MDRB

COVID-19 was significantly associated, before and after adjustment for predetermined confounders, with ICU-acquired BSI related to MDRB (adjusted cHR 2.65 (95%IC 1.25–5.59) for the whole ICU period) (Table 4). However, the proportional hazard assumption was not respected, with only a significant association for the period starting at day 15 after ICU admission (Figure 1).

### 3.4. Relationship between ICU-Acquired BSI Related to MDRB and Mortality, Length of ICU Stay, and Duration of Mechanical Ventilation

The cumulative incidence of ICU mortality, extubation alive (for the 659 patients with IMV), and ICU discharge alive are shown according to COVID-19 status in the Appendix A.

ICU-acquired BSI related to MDRB was significantly associated, before and after adjustment for predetermined confounders, with increased mortality. However, COVID-19 status had no significant impact on this relationship (P het = 0.94) (Figure 2).

ICU-acquired BSI related to MDRB was not significantly associated, before and after adjustment for predetermined confounders, with length of ICU stay or duration of mechanical ventilation. COVID-19 status had no significant impact on this relationship (P het 0.68 and 0.95, respectively) (Figure 2).

## 4. Discussion

The main results of our study are: ICU-acquired BSI related to MDRB was not common in our ICU (50 first episodes in 1320 patients hospitalized for >48 h (3.79%)). ESBL-Enterobacteriacae (46%) and carbapenem-resistant *A. baumannii* (30%) were the most commonly isolated MDRB. COVID-19 was significantly associated with a higher risk of ICU-acquired BSI related to MDRB for the whole study period. However, this relationship was only significant for the period starting at day 15 after ICU admission. ICU-acquired BSI related to MDRB was significantly associated with ICU mortality, although COVID-19 had no significant impact on this association. ICU-acquired BSI related to MDRB was not significantly associated with length of ICU stay or duration of mechanical ventilation, and COVID-19 had no significant impact on this association.

To our knowledge, our study is the first to specifically evaluate the relationship between COVID-19 and ICU-acquired BSI related to MDRB. Previous studies have reported an increased risk of BSI in COVID-19 patients compared with non-COVID-19 patients [17]. Further, a recent descriptive analysis of the Eurobact II study reported a higher incidence of difficult-to-treat Gram-negative bacteria among COVID-19 patients with BSI compared with non-COVID-19 patients with BSI (19.4% vs. 13%, *p* = 0.017) [18]. The higher risk of ICU-acquired BSI related to MDRB found by our study appeared starting at day 15 after ICU admission. This result is in line with the findings of Buetti et al., showing that the risk of BSI significantly increased in COVID-19 patients starting at day 7 after ICU admission [17]. Other studies reported low risk for early infections in COVID-19 patients, and higher risk for late-onset infections related to MDRB [8,9].

The duration of exposure to risk factors for BSI is a key point in the pathophysiology of these infections [30], and might explain the higher risk for ICU-acquired BSI related to MDRB starting at day 15 after ICU admission. Previous studies identified COVID-19 as an independent risk factor for ICU-acquired infections, including BSI [12,14,17]. Prolonged ICU stay, frequent antibiotic therapy, invasive devices, and corticosteroid use could explain, at least in part, the higher risk for BSI in COVID-19 patients [31,32,33,34]. However, a causal relationship between COVID-19 and ICU-acquired infection was suggested, and could be related to major immune dysregulation, often compared to the “cytokine storm” [35,36], post-aggressive immunoparalysis [37,38,39,40], and altered microbiota [41]. A recent large multicenter before-and-after study reported a higher risk for ICU-acquired infections related to MDRB in COVID-19 patients as compared to non-COVID-19 patients. The absence of a significant difference in ICU-acquired colonization related to MDRB in that study suggested a causal link between COVID-19 and ICU-acquired infection related to MDRB [42].

ICU-acquired BSI related to MDRB was significantly associated with mortality in our study, which is in line with previous findings [43,44]. In contrast with previous studies, no significant relationship was found between ICU-acquired BSI related to MDRB and the duration of mechanical ventilation or length of ICU stay. This is probably related to the relatively small number of patients with ICU-acquired BSI related to MDRB (*n* = 50). The absence of a significant impact of COVID-19 on mortality, duration of mechanical ventilation, and length of ICU stay could also be related to the limited number of patients in the subgroups. However, this was a secondary objective of our study and further studies are required to evaluate this point.

Our study has some limitations. First, it was retrospective and performed in a single center, which makes it difficult to generalize the results, especially since the occurrence of infections related to MDRB is highly dependent on local practices and ecology. However, the incidence of MDRB in the non-COVID group is comparable to that described in multicenter cohorts [2]. Similarly, the microorganisms found in our study are comparable to those described in the literature, both in non-COVID patients [3] and COVID patients [17,45], with a majority of Gram-negative bacilli, mainly *Acinetobacter baumannii*. However, some studies described a higher rate of *Enterococcus* infections [18,46], which was not the case in our center. Second, no data were collected on type of infections and antibiotics used before BSI related to MDRB. Third, no data were provided on the treatment of BSI related to MDRB or on the impact of different MDRB on outcomes. However, the small number of patients in different subgroups of MDRB preclude any valuable conclusion. Finally, our study did not evaluate the incidence of BSI related to bacteria other than MDRB.

## 5. Conclusions

In conclusion, the results of this retrospective single-center study suggest a higher risk for ICU-acquired BSI related to MDRB in COVID-19 patients as compared with non-COVID-19 patients, mainly during the period starting at day 15 after ICU admission. Further, COVID-19 had no significant impact on the relationship between ICU-acquired BSI related to MDRB and mortality, duration of mechanical ventilation, and length of ICU stay. Future studies are required to confirm our results.

## Figures and Tables

**Figure 1 antibiotics-12-01105-f001:**
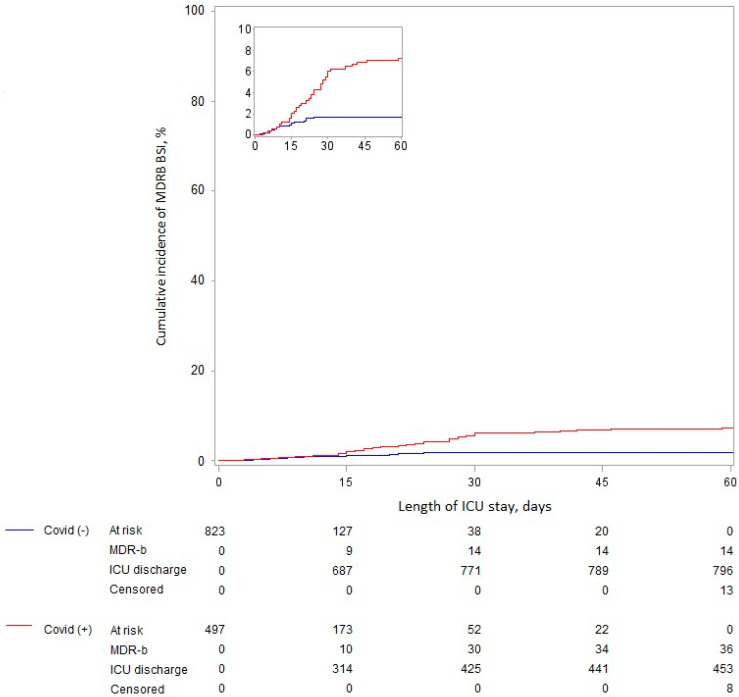
Cumulative incidence of ICU-acquired BSI related to MDRB according to COVID-19 status, considering death as a competing event.

**Figure 2 antibiotics-12-01105-f002:**
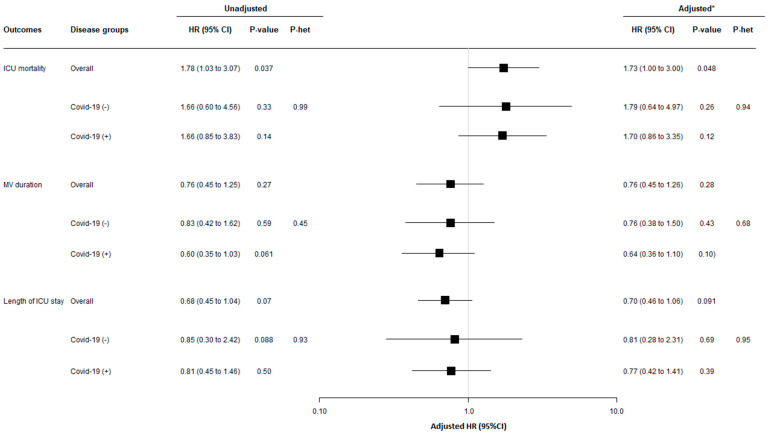
**Relationship between ICU-acquired BSI related to MDRB and 60-day outcomes, overall and according to study groups**. Mechanical ventilation was initiated in 485 patients (264 with COVID-19), and 45 ICU-acquired BSIs related to MDRB occurred during mechanical ventilation (34 with COVID-19). HRs were calculated using cause-specific proportional hazard models considering mortality as a competing event for mechanical ventilation and length of ICU stay, and considering ICU-discharge alive as a competing event for ICU mortality. ***** Adjusted HRs were calculated including age, SAPS II, immunosuppression, recent hospitalization, recent antibiotic treatment, colonization related to MDRB at ICU admission, and chronic dialysis, as pre-specified covariates in Cox’s models. The event of interest for survival is a pejorative event (death), whereas for MV duration and ICU length of stay, the event of interest is a positive event (extubation or discharge alive). Consequently, the detrimental effect of ICU-acquired BSI related to MDRB on each outcome was associated with a HR > 1 for ICU survival but with a HR < 1 for MV duration and ICU length of stay. P het indicates a Chi-squared test for heterogeneity in association with occurrence of MDRB and 60-day outcomes across study groups. Abbreviations: CI, confidence interval; HR, hazard ratio; ICU, intensive care unit; MV, mechanical ventilation.

**Table 1 antibiotics-12-01105-t001:** Patient characteristics at ICU admission.

	Overall	COVID-19		
	*n* = 1320	Yes *n* = 497	No *n* = 823	Standardized Difference %
Age (years)	62.0 (50.0–70.0)	65.0 (56.0–73.0)	60.0 (47.0–69.0)	36.6
Male gender	884 (67.0)	364 (73.2)	520 (63.2)	21.7
BMI (kg/m^2^) ^a^	27.4 (23.7–32.6)	29.4 (25.7–34.6)	26.1 (22.4–31.2)	53.4
SAPS II ^b^	39.0 (29.0–55.0)	38.0 (30.0–51.0)	40.0 (29.0–57.0)	−8.9
SOFA ^c^	4 (2–8)	4 (3–8)	4 (2–7)	4.5
**Comorbidities**				
Current smokers	367 (27.8)	77 (15.5)	290 (35.2)	−46.6
Alcohol consumption	216 (16.4)	39 (7.8)	177 (21.5)	−39.3
Diabetes mellitus	392 (29.7)	158 (31.8)	234 (28.4)	7.3
Hypertension	646 (48.9)	260 (52.3)	386 (46.9)	10.8
Ischemic heart disease	188 (14.2)	76 (15.3)	112 (13.6)	4.8
Heart failure	154 (11.7)	39 (7.8)	115 (14.0)	−19.8
Thromboembolic disease	86 (6.5)	27 (5.4)	59 (7.2)	−7.2
Lung disease	250 (18.9)	48 (9.7)	202 (24.5)	−40.3
Chronic obstructive pulmonary disease	177 (13.4)	48 (9.7)	129 (15.7)	−18.2
Hemodialysis	128 (9.7)	39 (7.8)	89 (10.8)	−10.2
Liver cirrhosis	56 (4.2)	6 (1.2)	50 (6.1)	−26.2
Hematological malignancy	88 (6.7)	22 (4.4)	66 (8.0)	−14.9
Solid organ transplantation	33 (2.5)	5 (1.0)	28 (3.4)	−16.4
Immunosuppressive treatment	146 (11.1)	31 (6.2)	115 (14.0)	−25.9
Corticosteroid use	108 (8.2)	30 (6.0)	78 (9.5)	−12.9
Solid neoplasia	136 (10.3)	31 (6.2)	105 (12.8)	−22.4
Metastatic neoplasia	53 (4.0)	11 (2.2)	42 (5.1)	−15.4
HIV	4 (0.3)	1 (0.2)	3 (0.4)	−3.1
Hospitalization < 3 months	352 (26.7)	87 (17.5)	265 (32.2)	−34.5
Surgery < 3 months	111 (8.4)	18 (3.6)	93 (11.3)	−29.5
Recent antibiotic treatment < 3 months	327 (24.8)	114 (22.9)	213 (25.9)	−6.9
MDRB colonization < 3 months	149 (11.3)	48 (9.7)	101 (12.3)	−8.4
**Location before ICU admission** ^d^				
Home	736 (57.5)	253/488 (51.8)	483/792 (61.0)	34.2
Hospital ward	368 (28.8)	131/488 (26.8)	237/792 (29.9)	
Another ICU	176 (13.8)	104/488 (21.3)	72/71,322 (9.1)	
**Type of admission**				48.3
Medical	1234 (93.5)	497 (100.0)	737 (89.6)	
Surgical	86 (6.5)	0 (0)	86 (10.4)	
**Cause of admission**				
Acute respiratory failure	796 (60.3)	475 (95.6)	321 (39.0)	151.1
ARDS	360 (27.3)	263 (52.9)	97 (11.8)	97.9
Congestive heart failure	35 (2.7)	1 (0.2)	34 (4.1)	−27.3
Septic shock	142 (10.8)	10 (2.0)	132 (16.0)	−50.5
Shock other than septic	72 (5.5)	2 (0.4)	70 (8.5)	−40.1
Sepsis	385 (29.2)	55 (11.1)	330 (40.1)	−70.6
Community-acquired pneumonia	130 (9.8)	38 (7.6)	92 (11.2)	−12.1
Healthcare-associated pneumonia	85 (6.4)	9 (1.8)	76 (9.2)	−32.9
Intra-abdominal infection	44 (3.3)	2 (0.4)	42 (5.1)	−29.0
Skin and soft tissue infection	65 (4.9)	4 (0.8)	61 (7.4)	−33.8
Neurological deficiency ^e^	164 (12.4)	8 (1.6)	156 (19.0)	−10.7
Acute kidney injury	91 (6.9)	9 (1.8)	82 (9.9)	2.2
Acute liver failure	17 (1.3)	0 (0)	17 (2.1)	7.2
Drug poisoning	99 (7.5)	28 (5.6)	71 (8.6)	−11.7
Hanging	28 (2.1)	0 (0)	28 (3.4)	−15.7
Cardiorespiratory arrest	62 (4.7)	2 (0.4)	60 (7.2)	−3.6
Other causes	111 (8.4)	12 (2.4)	98 (11.9)	−36.4

Data are presented as *n* (%) or median (interquartile range). Abbreviations: BMI, body mass index; SAPS II, Simplified Acute Physiology Score II; SOFA, Sepsis Related Organ Failure Assessment score; HIV, human immunodeficiency virus; ICU, intensive care unit; ARDS, acute respiratory distress syndrome; MDRB, multidrug-resistant bacteria. ^a^ 113 missing values (88 for COVID+, 25 for COVID−). ^b^ 3 missing values (3 for COVID−). ^c^ 80 missing values (32 for COVID+, 48 for COVID−). ^d^ 40 missing values (39 for COVID+, 1 for COVID−). ^e^ 2 missing values (1 for COVID +, 1 for COVID−)

**Table 2 antibiotics-12-01105-t002:** Patient characteristics during ICU stay.

	Overall	COVID-19		
	*n* = 1320	Yes *n* = 497	No *n* = 823	Standardized Difference, %
**Invasive devices and procedures**				
Central venous catheters ^a^	807 61.5)	297 (59.9)	510 (62.4)	−5.2
Arterial catheters ^b^	952 (73.2)	430 (87.0)	522 (64.7)	54.1
Dialysis catheters	140 (10.6)	59 (11.9)	81 (9.8)	6.5
Invasive mechanical ventilation	672 (50.9)	269 (54.1)	403 (49.0)	10.3
Duration (days) ^c^	8.0 (4.0–17.0)	14.0 (8.0–24.0)	6.0 (3.0–11.0)	91.3
Tracheostomy	80 (6.1)	51 (10.3)	29 (3.5)	26.8
Prone positioning	220 (16.7)	185 (37.2)	35 (4.3)	89.0
ECMO/ECLS	61 (4.6)	49 (9.9)	12 (1.5)	37.0
**Treatments**				
Parenteral nutrition	111 (8.4)	66 (13.3)	45 (5.5)	27.1
Transfusion	326 (24.7)	123 (24.7)	203 (24.7)	0.2
Antibiotics	1134 (85.9)	467 (93.9)	667 (81.0)	39.8
Steroids	619 (46.9)	376 (75.7)	243 (29.5)	104.2
Tocilizumab	15 (1.1)	11 (2.2)	4 (0.5)	15.0
Other immunosuppressive treatments	34 (2.6)	15 (3.0)	19 (2.3)	4.4

Data expressed as *n* (%) or median (interquartile range). Abbreviations: ECMO, extracorporeal membrane oxygenation; ECLS, extracorporeal life support; ICU, intensive care unit. ^a^ 7 missing values (1 for COVID+, 6 for COVID−). ^b^ 19 missing values (3 for COVID+, 16 for COVID−). ^c^ Calculated for 659 ventilated patients, 4 missing values (4 for COVID+).

**Table 3 antibiotics-12-01105-t003:** Characteristics of patients with ICU-acquired BSI related to MDRB.

	Overall	COVID-19	
	*n* = 50	Yes *n* = 36	No *n* = 14
**Type of MDRB BSI**			
ESBL Enterobacteriaceae	23 (46.0)	12 (33.3)	11 (71.4)
CR-Enterobacteriaceae	9 (18.0)	8 (22.2)	1 (7.1)
CR-*Acinetobacter baumannii*	15 (30.0)	13 (36.1)	2 (14.2)
MDR *Pseudomonas aeruginosa*	2 (4.0))	2 (5.5)	0 (0)
MRSA	1 (2.0)	1 (2.7)	0 (0)
**Source of MDRB BSI**			
Catheter-related	12 (24.0)	10 (27.8)	2 (14.3)
Respiratory tract	26 (52.0)	20 (55.6)	6 (42.9)
Unknown	9 (18.0)	4 (11.1)	5 (35.7)
Intra-abdominal	1 (2.0)	1 (2.8)	0 (0)
Urinary tract	1 (2.0)	1 (2.8)	0 (0)
Multiple	1 (2.0)	0 (0)	1 (7.1)

Data expressed as *n* (%). Abbreviations: MDRB, multidrug-resistant bacteria; BSI, bloodstream infection; ESBL, extended-spectrum beta-lactamase; CR: carbapenem-resistant; MRSA, methicillin-resistant *Staphylococcus aureus.*

**Table 4 antibiotics-12-01105-t004:** Unadjusted and adjusted effect size of the relationship between COVID-19 status and the incidence of ICU-acquired BSI related to MDRB.

	COVID-19		Unadjusted Analysis		Adjusted Analysis ^c^	
	No (*n* = 823)	Yes (*n* = 497)	cHR (95%CI)	*p*-Value	cHR (95%CI)	*p*-Value
Overall period ^a^	14/823 (1.7)	36/497 (7.2)	2.22 (1.18 to 4.13)	0.012	2.65 (1.25 to 5.59)	0.011
0 to 14 days ^b^	8/823 (1.0)	8/497 (1.6)	1.11 (0.41 to 2.97)	0.84	1.40 (0.49 to 3.99)	0.53
15 to 60 days ^b^	6/137 (4.2)	28/190 (14.8)	3.52 (1.45 to 8.52)	0.005	4.35 (1.58 to 11.90)	0.004

Values are number of events/number of patients at risk (cumulative incidence %), otherwise as indicated. Abbreviations: cHR, cause-specific hazard ratio; MDRB, multidrug-resistant bacteria. ^a^ Capped at 60 days. ^b^ To accommodate the non-proportional hazard assumption of the effect of COVID-19 status, the association was further modeled using time-dependent coefficients. ^c^ Adjusted for pre-specified baseline confounders (age, SAPS II, immunosuppression, recent hospitalization, recent antibiotic treatment, colonization related to MDRB at ICU admission, and chronic dialysis).

## Data Availability

Data are available upon request to the corresponding author.

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
