# Peer review of "Relationship between COVID-19 and ICU-Acquired Bloodstream Infections Related to Multidrug-Resistant Bacteria"

_antibiotics, 2023, doi:10.3390/antibiotics12071105_

Round 1
Reviewer 1 Report
The Authors tried to describe the relationship between COVID-19 and mrdo ICU-acquired BSI.
Although the subject of the paper could be interesting, the whole manuscript appears to be incomplete and not fully adequate for publication.
In particular:
- references should be restructed from the begininng: the order did not respect number progression, a lot of references were inserted as "authors, doi..." and so on
- it is not clear why the authors did not provide a statistical analysis of data shown in table 1 and 2. As a consequence, they can not describe those data as "comparable" or similar.
- line 73: Authors stated that data were collected prospectively but they previously defined the study as retrospective
- in the whole discussion section a lot of references were missed (see the word "ref" repeated several times at the end of multiple phrases)
- Authors should specify what they intended as confunders in the adjected analysis
- A known risk factor for ICU acquired infection and BSI is "length of ICU stay" (see 1- Giacobbe DR et al, doi 10.1111/eci.13319 and 2- Gouel-Cheron et al doi 10.1097/CCM.0000000000005662) so it appears mandatory to to evaluate this item in the analysis
- conclusions do not appear completely consistent with the results
The paper does not appear adequate for publication. A complete rewriting and a new analysis seem mandatory.
Reviewer 2 Report
I read with interest the paper by Piantoni et al. They evaluated the impact of bloodstream infection caused by multidrug resistant bacteria in a cohort of COVID-19 critically ill patients over 2020. The topic is of interest for a general readership to understand the burden of multidrug resistance as COVID-19 legacy. However, I have considerations to bring to the attention of the authors and would like to point out how the work in certain parts appears to be presented without due care and the text appears to be in draft format. I would advise the authors from the outset to be more careful when submitting their work to international journals, respecting everyone's task.
Introduction:
Lines 31-33: Please provide reference.
Line 40: Please report the reference in the right format.
Line 45: Please report the reference in the right format.
Line 47: Please report the reference in the right format.
Methods:
The methods section is not very robust, the definitions and microbiological methods are approximate:
1. Do the author think that a positive blood culture is sufficient to define an episode of bloodstream infection? If so, please provide reference.
2. What kind of RT-PCR did the authors use?
3. Lines 69-72: I think the authors should give a definition of multidrug resistance and only in the results make explicit which types of multidrug-resistant bacteria were found and characterise them by mechanism of resistance.
4. What kind of microbiological tests did the authors use to obtain the antimicrobial susceptibility results and resistance patterns?
5. Lines 72-72: This sounds to be reported in the results section and not in the methods.
6. Lines 95-97: Please provide the reference in the right format.
The results are also interesting, but I have to admit that the carelessness of the text (in the discussion there are poorly reported references and sentences in French, see lines 300-301) make it very difficult to read, so I advise the authors to take their time to revise the work and resubmit it in an appropriate form.
Reviewer 3 Report
- The introduction is quite superficial. Authors should deeply described the correlation of COVID-19 and MDR infections. Several papers assessed this topic and should be included in this section and also in the references (Lai CC, Co-infections among patients with COVID-19: The need for combination therapy with nonanti-SARS-CoV-2 agents? J. Microbiol. Immunol. 2020. Giacobbe DR, Bloodstream infections in critically ill patients with COVID-19. Eur. J. Clin. Investig. 2020. Sogaard KK,Community-acquired and hospital-acquired respiratory tract infection and bloodstream infection in patients hospitalized with COVID-19 pneumonia. J. Intensive Care 2021. Bongiovanni M, Pseudomonas aeruginosa Bloodstream Infections in SARS-CoV-2 Infected Patients: A Systematic Review, JCM 2023, etc).
- The first case of COVID in France was diagnosed at January 24th 2020. I guess it's not methodologically correct that the study includes patients starting from the beginning of 2020. Probably starting the analysis from February could be more appropriate.
- The patients admitted in ICU for sepsis or infections at baseline should be described deeply. Which kind of bacteria they had? With which antibiotics were treated? The occurrence of MDR infections was associated with empiric large-spectrum antibiotic treatment at baseline?
- Authors should analyze data according to the different waves of COVID-19 infections occurred in 2020 (at least March-May 2020 vs October-December 2020) to discuss if differences were present in thiese timeframes.
- The antibiotic treatments used for MDR infections should be described and discuss, as well as the specific outcomes of patients according to the isolated MDR bacteria
- The reference section has some mistakes (i.e. the paper cited by Conway-Morris is referenced as number 38, but it's at the beginning of the paper and the total number of reference is 31). Please clarify
English should be revised
Round 2
Reviewer 1 Report
The Authors described BSI due to MDRO in severe COVID-19 patients.
The queries of the previous review were adequately answer and the paper significantly improved.
Author Response
We thank the reviewer for his/her comment.
Reviewer 2 Report
I would like to thank the authors for finally revising their work and bringing it up to international journal standards.
Methods:
1) What method for bacterial identification did the authors use?
2) Which clinical breakpoints did the authors use?
Results:
3) The authors stated that their study was aimed at comparing the incidence of ICU-acquired MDR-BSI in COVID-19 and non-COVID-19 patients and evaluating the effect of COVID-19 on the relationship between ICU-acquired MDR-BSI and mortality, duration of invasive mechanical ventilation, and length of ICU stay.
I think that for these purposes, the overall number of patients to be considered is 50 (36 COVID-19 and 14 non-COVID-19), so I consider the data in table 1 and 2 to be absolutely not useful.
4) The authors said they used an immunochromatographic assay to characterise carbapenemase-producing strains. What were the results of this characterisation? I think indicating the type of carbapenemase is relevant because there are treatment regimes with different efficacy depending on the type diagnosed.
4) Lines 194-196: “A higher proportion of carbapenem-resistant Enterobacteriaceae, carbapenem-resistant Acinetobacter baumannii, and multidrug-resistant Pseudomonas aeruginosa was found in the COVID-19 group”. I do not think the numbers (8 vs 1, 13 vs 2, 2 vs 0) are such that any conclusions can be drawn. Do the authors think so?
5) I think the authors should revise the work in light of these considerations and calibrate it to the study population (n=50). In my opinion, the small numbers and numerous limitations partially stated in lines 305 to 317 make the work difficult to publish as original article in a journal with a high impact factor.
